# Duckweed hosts a taxonomically similar bacterial assemblage as the terrestrial leaf microbiome

Kenneth Acosta[1], Jenny Xu[1], Sarah Gilbert[1], Elizabeth Denison[2], Thomas Brinkman[1], Sarah Lebeis[2], Eric Lam[1]*

1 Department of Plant Biology, Rutgers the State University of New Jersey, New Brunswick, New Jersey, United States of America, 2 Department of Microbiology, University of Tennessee, Knoxville, Tennessee, United States of America

* ericl89@hotmail.com

**Data Availability Statement:** Duckweed barcode sequences and 16rDNA amplicon libraries are available under the NCBI Bioproject ID PRJNA561628. All QIIME 2 and R code is available

## Abstract

Culture-independent characterization of microbial communities associated with popular plant model systems have increased our understanding of the plant microbiome. However, the integration of other model systems, such as duckweed, could facilitate our understanding of plant microbiota assembly and evolution. Duckweeds are floating aquatic plants with many characteristics, including small size and reduced plant architecture, that suggest their use as a facile model system for plant microbiome studies. Here, we investigated the structure and assembly of the duckweed bacterial microbiome. First, a culture-independent survey of the duckweed bacterial microbiome from different locations in New Jersey revealed similar phylogenetic profiles. These studies showed that Proteobacteria is a dominant phylum in the duckweed bacterial microbiome. To observe the assembly dynamics of the duckweed bacterial community, we inoculated quasi-gnotobiotic duckweed with wastewater effluent from a municipal wastewater treatment plant. Our results revealed that duckweed strongly shapes its bacterial microbiome and forms distinct associations with bacterial community members from the initial inoculum. Additionally, these inoculation studies showed the bacterial communities of different duckweed species were similar in taxa composition and abundance. Analysis across the different duckweed bacterial communities collected in this study identified a set of "core" bacterial taxa consistently present on duckweed irrespective of the locale and context. Furthermore, comparison of the duckweed bacterial community to that of rice and Arabidopsis revealed a conserved taxonomic structure between the duckweed microbiome and the terrestrial leaf microbiome. Our results suggest that duckweeds utilize similar bacterial community assembly principles as those found in terrestrial plants and indicate a highly conserved structuring effect of leaf tissue on the plant microbiome.

at https://github.com/kenscripts/duckweed_
microbiome.

**Funding:** Duckweed research in EL laboratory was
supported in part by a grant from the Department
of Energy (DE-SC0018244) and a Hatch project
(#12116) from the New Jersey Agricultural
Experiment Station at Rutgers University. SL
acknowledges start-up funds provided to her from
the University of Tennessee, Knoxville. The funders
had no role in study design, data collection and
analysis, decision to publish, or preparation of the
manuscript.

**Competing interests:** The authors have declared
that no competing interests exist.

## Introduction

Terrestrial plants harbor a multitude of microorganisms that can confer fitness advantages
either through plant growth promotion or disease protection [1]. Beneficial bacterial members
with plant growth promoting ability are commonly isolated and applied to sustainably
improve crop yield. However, plant-growth promoting bacteria (PGPB) are often not success-
ful in the field mainly due to the inability of PGPB to form stable associations with plant hosts
over time [2]. A greater understanding of the interactions occurring between plant hosts and
their myriad associated microorganisms, as well as between the microbes themselves, will be
necessary for improving PGPB field efficacy [3,4].

To improve our understanding of these interactions, a number of culture-independent
studies, using next-generation sequencing technologies, have been conducted on terrestrial
plants to characterize the plant microbiome. These analyses showed a consistent assemblage of
bacteria from the phyla Actinobacteria, Bacteroidetes, Firmicutes, and Proteobacteria on both
roots and leaves of terrestrial plants [5–7]. Moreover, a number of factors can affect the com-
position of the terrestrial plant-associated microbial community including plant compartment
and soil inoculum as strong determinants as well as plant genotype, developmental stage, and
cultivation practice as relatively minor determinants [8–10]. In addition to culture-indepen-
dent studies, reductionist approaches using gnotobiotic plants, culture collections of plant
microbiota, and sterile soil matrix have begun to resolve the complexity of interactions occur-
ring in the plant microbiome [11,12]. While these approaches have mainly used terrestrial
plants, implementation of other model systems could facilitate our understanding of the plant
microbiome and the mechanisms that are involved in shaping its population structure.

Duckweed possesses several desirable characteristics that warrant its use as a model system
to study plant microbial communities. Duckweeds are aquatic plants that belong to the family
Lemnaceae and are composed of 5 genera and 37 species [13]. Duckweed has been used in
many ecotoxicological and phytoremediation studies and there is growing interest in duck-
weed as biofuel, animal feedstock, and food [14,15]. The recent development of duckweed
genomic resources and molecular tools have positioned duckweed as a model system for sev-
eral aspects of plant biology [16–19]. Duckweed has a simple body architecture consisting of
mainly leaf-like structures termed fronds that float on the water surface while roots are only
found in some species as simple roots with no lateral branching or root hairs [13]. Duckweed
is able to maintain this simple architecture throughout its life cycle since it mainly propagates
clonally in the laboratory. Duckweed is only a few millimeters in size and one of the fastest
growing plant species which enable economy of space and time with their study [20]. These
traits can be leveraged to facilitate high-throughput microbiome studies as they were previ-
ously exploited to develop screens for microbial pathogenesis [21,22]. More importantly, its
aquatic habitat is relatively homogenous compared to soil and allows for straightforward sam-
pling in addition to accurate and robust measurements. As an angiosperm, duckweed may
provide additional information on the evolution of plant microbiomes since the monocot
ancestors for this plant family transitioned from terrestrial environments back to an aquatic
lifestyle about 100 Mya [23].

Here, we analyzed the composition and assembly of the duckweed bacterial microbiome
using 16S rRNA gene community profiling. Proteobacteria were the most abundant taxa in
bacterial communities collected from both wild duckweed and duckweed inoculated with
wastewater treatment effluent. Diversity analysis and differential abundance testing of duck-
weed bacterial communities assembled from wastewater inoculum showed that particular
duckweed-bacterial associations were selected from the environment. Similar bacterial com-
munities were found on wild duckweed collected from different sites while inoculation studies

showed similar bacterial communities assembled onto different duckweed species. By combining the different duckweed bacterial communities generated in this and other studies, we identified a set of "core" bacterial taxa consistently associated with duckweed in moderate abundance. Comparison of the duckweed bacterial community to that of rice and Arabidopsis revealed a taxonomically similar leaf bacterial microbiome. Together these data suggest similar structuring principles govern the assembly of duckweed and terrestrial plant microbiomes with a conserved leaf organizational effect on the plant microbiome. The data presented in this work should facilitate the development of experimental approaches to understand plant microbial community establishment and the application of stable "core" microbiota for improving performance of duckweed-based applications.

## Materials and methods

### Survey of the duckweed bacterial community

**Sample collection.** To survey the duckweed bacterial microbiome both duckweed and ambient water samples were collected from two ponds located in New Brunswick, NJ (Passion Puddle) and Great Meadows, NJ (Caldwell House) (Fig 1).

Ambient water samples were passed through sterilized Miracloth to remove solids. Ambient water was then filtered through a 0.2 um 150 mL Nalgene rapid filter unit (ThermoScientific, Catalog No. 125–0020) to capture the microbial community. Filter membranes were excised, placed in 5 mL centrifuge tubes, flash frozen in liquid nitrogen, and stored until processing (protocols.io DOI: dx.doi.org/10.17504/protocols.io.98zh9x6).

Duckweed tissue was first separated from other solids. Duckweed samples were then washed with salt and detergent (1X PBS, 0.1% Triton X-100, 0.5 mM MgSO4, 1 mM CaCl2) and rinsed twice with water. Duckweed tissue was placed in 5 mL centrifuge tubes and flash frozen in liquid nitrogen. Samples were stored at -80˚C until processing (protocols.io DOI: dx. doi.org/10.17504/protocols.io.98zh9x6).

**Duckweed genotyping.** Duckweed collected from the environment were genotyped using two barcodes: *atpF-atpH* (5'-ACTCGCACACACTCCCTTTCC-3' and 5'-GCTTTTATGGAA GCTTTAACAAT-3') and *psbK-psbI* (5'-TTAGCATTTGTTTGGCAAG-3' and 5'-AAAGT TTGAGAGTAAGCAT-3') as suggested in [24]. Different duckweed samples were surface

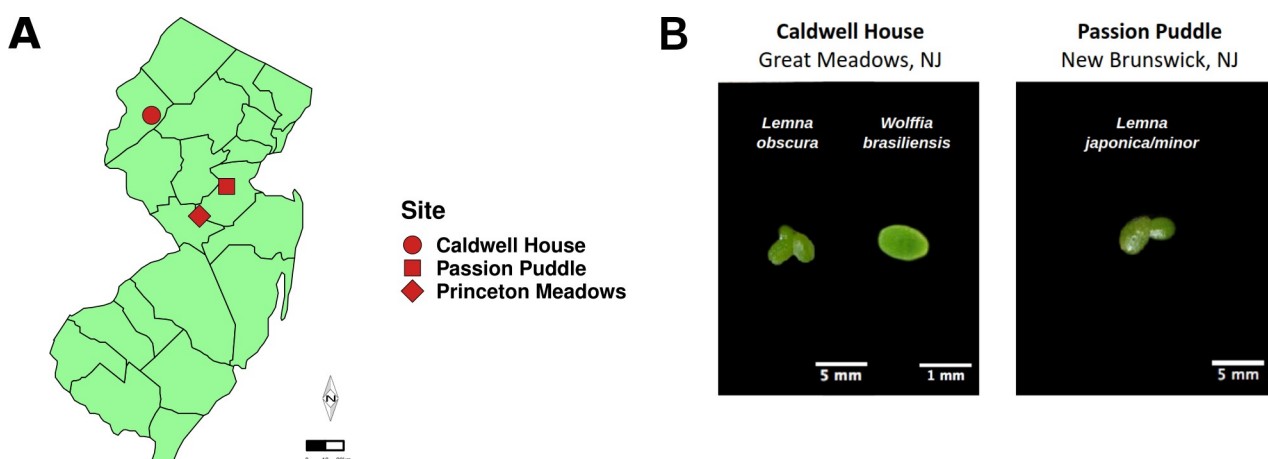

**Fig 1. Sample collection sites.** (**A**) Map of New Jersey, USA depicting sample collection sites. Duckweed and ambient water samples were collected from Caldwell House and Passion Puddle sites. Wastewater was collected from United Water Princeton Meadows treatment facility to use as an inoculum for assembly studies. (**B**) Representative images of duckweed collected from Caldwell House and Passion Puddle sites.

sterilized using 10% bleach, washed with 2% sodium thiosulfate followed by water, and cultured on media containing 0.5X Schenk and Hildebrandt (SH) (Phytotechnology Laboratories, Catalog No. S816), 0.1% sucrose, and 100 ug/mL cefotaxime. Duckweed are maintained and registered at the Rutgers Duckweed Stock Cooperative.

## Bacterial community inoculation studies

**Duckweed tissue propagation.**    *Lemna minor* 370–5576 (Lm5576) and *Spirodela polyrhiza* 432–9509 (Sp9509) were obtained from the RDSC. Duckweed tissue was first propagated in 50 mL baby food jars containing sterile growth media (0.5X SH and 0.1% sucrose at pH 7), then in 200 mL of growth media, and lastly in 400 mL of growth media for approximately two weeks at a time.

**Sample inoculation.**    For inoculation studies, wastewater from the United Water Princeton Meadows wastewater treatment facility in Plainsboro, NJ was collected after secondary clarification in the years 2015 (year 1) and 2016 (year 2). A written agreement made between United Water Princeton Meadows Inc. and Rutgers University granted us permission to collect wastewater samples from this site and use them for this study.

For the Princeton Meadows year 1 study, 75 mL of Princeton Meadows wastewater, collected in 2015, was inoculated with approximately 200 mg fresh weight Sp9509. Samples were collected at 0, 5, and 10 days at 26°C and a photo-peroid of 16 hr day and 8 hr night. Samples were harvested using 0.2 um Nalgene rapid-flow filter units. Miracloth was overlaid on the filter unit to capture duckweed tissue. Duckweed tissue was transferred to 5 mL centrifuge tubes and washed with salt and detergent followed by sterile water two times. Tissue was then flash frozen with liquid nitrogen and stored at -80°C until processing. Wastewater and ambient wastewater samples were filtered through 0.2 um membrane to capture the microbial community. Membranes were excised, placed in 5 mL centrifuge tubes, flash frozen in liquid nitrogen, and stored at -80°C until processing.

In the Princeton Meadows year 2 study 50 mL of Princeton Meadows wastewater, collected in 2016, was inoculated with either Lm5576 or Sp9509. Princeton Meadows samples were harvested at 2 and 7 days. Duckweed tissue was either treated with water only or with salt and detergent followed by two water washes. Tissue and ambient wastewater samples were processed as mentioned above.

## DNA isolation and library preparation

Frozen duckweed tissue was homogenized for 15 minutes at 1500 RPM in the Geno/Grinder 2010 (SPEX SamplePrep) with approximately 20 sterile garnet beads (0.7 mm, Qiagen). DNA was extracted using PowerSoil DNA Extraction Kit (MoBio) and stored at -80°C. For water samples, DNA was extracted from filters using the PowerWater DNA Extraction Kit (MoBiol). All DNA sample concentrations were quantified using the PicoGreen dsDNA kit (Invitrogen). To ensure that each sample contained amplifiable DNA, all samples had a 16S rRNA gene PCR performed using 515F and 806R primers as quality control before amplicon library construction.

Library preparation and sequencing was performed at the Joint Genome Institute as a part of the Community Science Program (Department of Energy, CSP project # DE-SC0018244). For library preparation, 25 µL reactions contained 11.4 µL PCR grade water, 1 µL BSA (10 mg/ml), 10 µL 5PRIME HotMaster Mix, 0.5 µL of 16S rRNA gene primers (10 µM), 1µL of DNA (10 ng/µL), and 0.3 µL (100 µM) of a mixture of two Peptide Nucleic Acids (PNA) one for blocking plant mitochondrial sequences and one for blocking plant plastid sequences [25]. For each sample, reactions were run in triplicate. All samples were amplified with the primers

515F-Y (5'-GTGYCAGCMGCCGCGGTAA-3') and 926R (5'-CCGYCAATTYMTTTRAGT TT-3') [26]. The thermocycler settings were: 3 minutes at 93°C, with 30 cycles of 94°C for 45 seconds, 78°C for 10 seconds (for PNA annealing), 50°C for 60 seconds, 72°C for 90 seconds and a 10 minute final extension at 72°C. Triplicate samples were combined and run on a 1% agarose gel to confirm PCR success and cleaned using Agencourt AMpure XBeads (Beckman Coulture) in a 1:1 ratio of beads to product according to the protocol specified in Illumina's 16S Metagenomic Sequencing Library Preparation. Secondary PCR to index each sample with unique adapters was performed after cleaning. Reactions for Index PCR consisted of 25 µL of 10 µL 5PRIME HotMaster Mix, 11.4 µL of sterile PCR grade water, 0.5 µL of both Nextera XT Index Forward and Reverse primer (JGI primers), 5 µL of cleaned DNA, and 0.3 µL (100 µM) of the 2 PNA mixture. The thermocycler settings were 94°C for 3 minutes, with 8 cycles of 94°C for 30 seconds, 78°C for 10 seconds (for PNA annealing), 50°C for 30 seconds, 72°C for 30 seconds and 5 minute final extension at 72°C. Indexing PCR success were visualized on 1% agarose gels and samples were cleaned again according to the same magnetic bead based protocol from Illumina. After the final clean up, the DNA concentration of all samples were quantified using a PicoGreen Assay (Invitrogen) and pooled equally according to their DNA concentration. The library was then processed at the Joint Genome Institute. They were first run on a Bioanalyzer High Sensitivity Chip (Agilent Technologies) to quantify concentration and confirm amplicon size then sequenced using Version 2, 300 cycle (2 X 275) kit on the Illumina MiSeq platform.

## Microbiome bioinformatics

**Creating feature tables and classifiers using QIIME 2.** For each experiment raw sequences were processed using QIIME 2 (q2) version 2018.6 [27]. Sequences were imported and demultiplexed using the SingleEndFastqManifestPhred33 Fastq manifest format. Quality control was performed and feature tables containing counts for the different amplicon sequence variants (ASVs) were produced using the q2-dada2 plugin [28].

ASV taxonomic classifiers were generated using the Greengenes 13_8 99% OTUs reference database and the q2-feature-classifier plugin with classify-consenus-blast as the classification method [29,30]. The classify-consensus-blast method was chosen due to its ability to classify a large percentage of reads (S1 Fig).

Feature tables were filtered for unclassified, mitochondria, chloroplast, and low frequency (> 1 read) ASVs.

**Diversity, taxonomic, and differential abundance analysis.** ASVs were aligned using the q2-alignment plugin [31] and phylogenies were constructed using the q2-phylogeny fasttree plugin [32]. The q2-diversity plugin was then used to rarefy each feature table and calculate the number of observed ASVs and Faith's phylogenetic diversity [33]. Unweighted UniFrac, weighted UniFrac, Jaccard, and Bray-Curtis distance metrics were also generated. The generalized UniFrac distance metric was produced using the GUniFrac R package [34]. The q2-taxa barplot plugin was used to determine the taxonomic composition of feature tables at the Phylum and Family levels. Differential abundance testing was conducted using ALDEx2 [35].

**Comparative analysis of duckweed and terrestrial plant microbiomes.** To compare the duckweed bacterial microbiome to terrestrial plant bacterial microbiomes, we gathered bacterial community data from two *Arabidopsis thaliana*, hereafter Arabidopsis, studies [36,37] comprising 48 root and 20 leaf samples and two rice studies [10, 38] comprising 126 root samples and 18 leaf samples (S1 and S2 Files). Different plant compartments were combined (S2 File).

### Data visualization, statistics, code, and data availability

QIIME 2 artifacts were exported and data visualizations were created using R version 3.6.0. Statistics were performed using R where appropriate. All QIIME 2 and R code along with manifest, metadata, input, and output files are available at: https://github.com/kenscripts/duckweed_microbiome. Amplicon libraries and duckweed barcode sequences have been uploaded under the NCBI BioProject ID PRJNA561628.

## Results

### Survey of the duckweed bacterial microbiome

To survey the composition of the duckweed bacterial microbiome, both duckweed and surrounding (ambient) water samples were collected from two residential ponds in New Jersey during the summer months (Fig 1). A two-barcode strategy was used to identify duckweed species [24]. Duckweed collected from the Caldwell House site was identified as *Wolffia brasiliensis* and duckweed from Passion Puddle was identified as *Lemna minor/japonica*. A much smaller quantity (< 1% in biomass) of *Lemna obscura* was also found in the Caldwell House site. To examine the bacterial community of samples, genomic DNA was extracted and the V4 region of the 16S rRNA gene was amplified and sequenced. Entire duckweed plants were processed due to the small size of duckweed (S3 File).

Samples were rarefied to a depth of 3664 reads to assess bacterial community diversity. At this sampling depth maximum diversity was captured in duckweed-associated bacterial (DAB) communities but some diversity was lost in ambient water communities (S2 Fig). DAB communities from both Caldwell House and Passion Puddle contained fewer amplicon sequence variants (ASVs) and lower phylogenetic diversity compared to ambient water bacterial communities (Fig 2). Principal coordinate analysis (PCoA) of unweighted and generalized UniFrac distances were performed to compare the diversity between DAB and ambient water bacterial communities. The unweighted UniFrac (UUF) distance metric does not consider taxa abundance and is sensitive to changes in taxa composition while the generalized UniFrac (GUF) distance is able to detect changes in abundance among moderately and highly abundant taxa [34]. PCoA results show DAB communities were significantly different from the ambient water bacterial communities (Fig 2 and S1 Table). While location had a noticeable effect on ambient water community structure, DAB communities from both locations were similar in composition (Fig 2 and S1 Table). Together these data suggest that duckweed hosts a conserved bacterial community that is distinct and less diverse than the surrounding water community.

We analyzed the taxa composition of Caldwell House and Passion Puddle bacterial communities to identify what kind of bacterial phyla, families, and genera associate with duckweed. Proteobacteria was the dominant bacterial phylum in the duckweed microbiome (82% and 90% in Caldwell House and Passion Puddle respectively) in contrast to the ambient water microbiome (47% in both Caldwell House and Passion Puddle) (Fig 3 and S4 File). While the Comamonadaceae family was the most abundant family in both Caldwell House and Passion Puddle DAB communities (24% and 25% average relative abundance respectively) some differences were observed in other DAB families between locations (Fig 3 and S4 File). At the genus level, *Rhodobacter*, *Agrobacterium*, *Hydrogenophaga*, *Bacillus*, and *Novosphingobium* were the top 5 most abundant genera found in the Caldwell House DAB community while *Rhizobium*, *Dechloromonas*, *Sphingomonas*, *Agrobacterium*, and *Sulfurospirillum* were the most abundant genera found in the Passion Puddle DAB community (S4 File). These data show that Proteobacteria is the major taxa constituent of the DAB community.

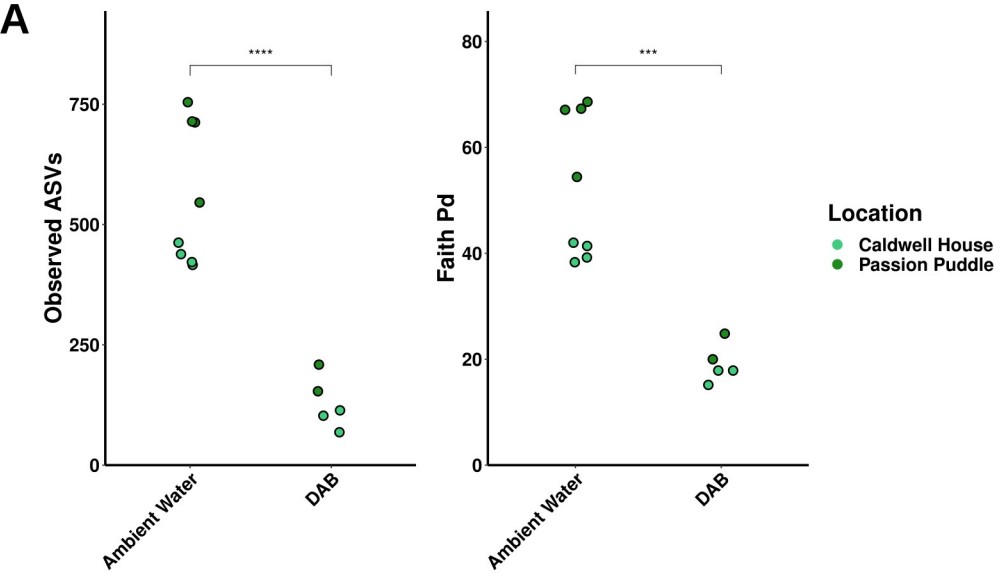

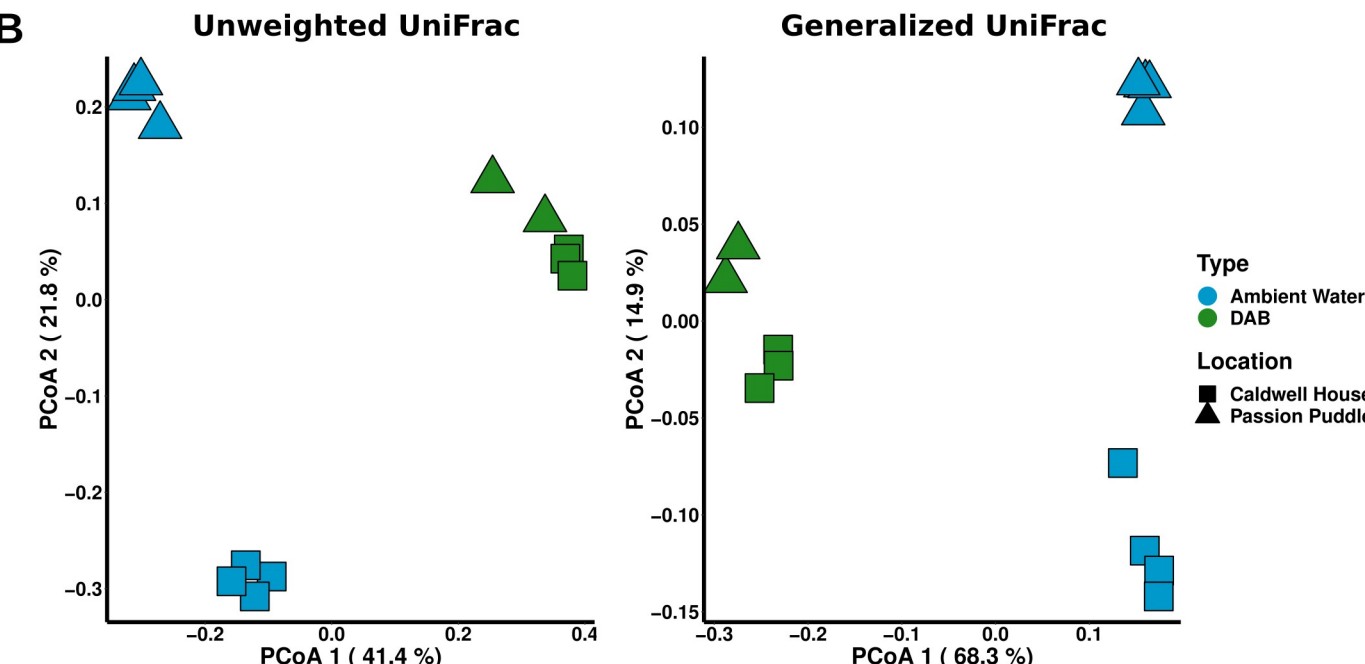

**Fig 2. Duckweed hosts a conserved bacterial community that is distinct from the surrounding water bacterial community. (A)** The total number of ASVs observed and assessment of the phylogenetic diversity using Faith's PD phylogenetic diversity index for DAB and ambient water communities from Caldwell House and Passion Puddle sites. Wilcoxon rank sum test was used for comparison of ASVs and Faith's PD index (p-value < 0.05 = "*", p-value < 0.01 = "**", p-value < 0.001 = "***", p-value < 0.0001 = "****"). **(B)** Principal coordinate analysis of DAB and ambient water bacterial communities from Caldwell House and Passion Puddle sites using unweighted UniFrac and generalized UniFrac distances.

We tested for differential member abundance between communities to verify differences. Unique methods have been developed for differential abundance testing in order to deal with the constraints imposed by compositional data such as that from 16S rRNA amplicon sequencing [39]. ALDEx2 was selected for differential abundance analysis because of its low false discovery rate and simplicity [40]. ALDEx2 transforms counts or member abundances into a distribution of centered-log ratios (clr) where the abundance of each member in a sample is

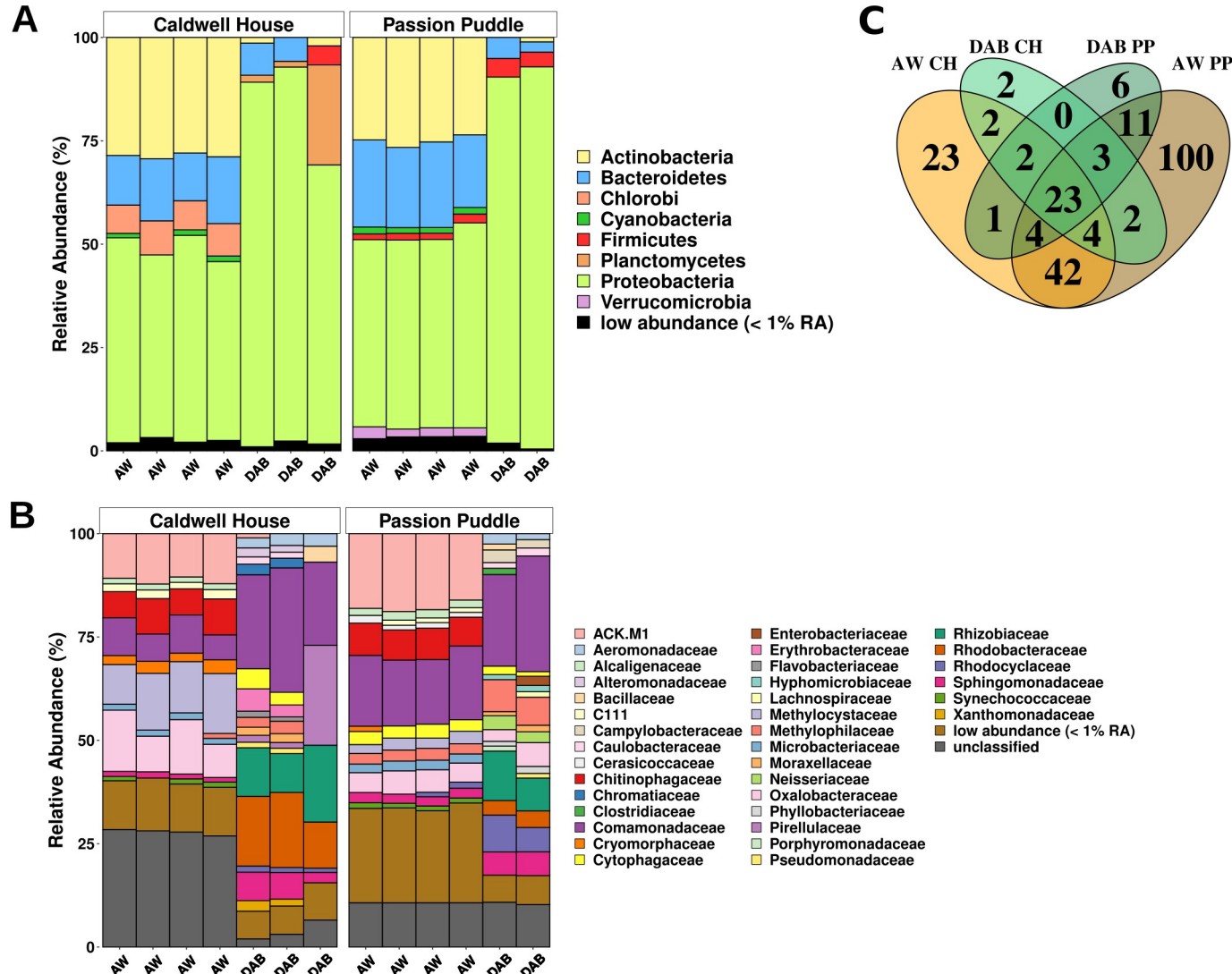

**Fig 3. Proteobacteria is the major constituent of the duckweed bacterial microbiome. (A)** Phylum, **(B)** family, and **(C)** genus composition of DAB and ambient water (AW) bacterial communities from Caldwell House (CH) and Passion Puddle (PP) sites.

compared to the sample's geometric mean. Those members with an average abundance have a value close to 0. Its output also includes effect size and significance testing to identify features that are different between groups. Beta-diversity analysis of duckweed collected from different sites indicated only certain bacteria associate with duckweed from the environment (Fig 2). Furthermore, a total of 101 bacterial genera were found in the Caldwell House ambient water but only 31 of these bacterial genera were found on duckweed while Passion Puddle ambient water contained 189 bacterial genera and only 41 were found on duckweed (Fig 3). To verify these differences, differential abundance testing was used to calculate which bacterial taxa were significantly enriched in DAB communities compared to the ambient water communities. ALDEx2 calculated 7 bacterial genera were significantly different in abundance between Caldwell House DAB and ambient water bacterial communities and 7 bacterial genera were all significantly enriched in the Passion Puddle DAB community compared to the ambient water (S3 Fig and S5 File). These DAB-enriched bacterial taxa were different between locations (S3

Fig). Diversity analyses of Caldwell House and Passion Puddle communities suggested a conserved DAB community between locations (Fig 2). Taxa analysis of shared and specific bacterial genera between locations revealed Caldwell House and Passion Puddle duckweed shared 28 bacterial genera while 10 bacterial genera were specific to Caldwell House duckweed and 22 bacterial genera were specific to Passion Puddle duckweed (Fig 3 and S4 File). Bacteria found in duckweed at one site and not found in another could be bacteria that are specific to that location. Of the 22 Passion Puddle-specific duckweed bacterial genera, 11 were specific to Passion Puddle ambient water (Fig 3 and S4 File). This included the genus *Rhizobium* which held the highest mean relative abundance among Passion Puddle duckweed-associated bacteria (S4 File). Alternatively, bacteria associated with duckweed in one location and not the other could be because the bacteria may not be captured by sequencing (low abundance bacteria), could represent random associations (found in only some samples), or may have been acquired from the phyllosphere at that particular location (abundant bacteria) (S4 File). However, ALDEx2 did not find any significant differences (adjusted Welch's t-test, $p < 0.05$) in abundance between the Caldwell House and Passion Puddle DAB communities but did find some differences between Caldwell House and Passion Puddle ambient water communities (S5 File). Together, these results suggest that a conserved bacterial community, composed mostly of Proteobacteria taxa, forms on wild duckweed collected from different environments.

## Assembly dynamics of the duckweed bacterial community

**Princeton Meadows year 1 study.** Two studies were conducted to investigate determinants of DAB community assembly. In the first study, we inoculated surface-sterilized *Spirodela polyrhiza* 9509 (Sp9509) with municipal wastewater effluent collected from Princeton Meadows in 2015 (Fig 1). A total of 31 samples were collected encompassing three types of bacterial communities: wastewater bacterial community not co-cultured with Sp9509 (WW), ambient wastewater bacterial community co-cultured with Sp9509 (AWW), and the bacterial community that assembled onto Sp9509 from the wastewater inoculum (WWDAB) (S3 File). Samples were collected at 0, 5, and 10 days post inoculation to determine if bacterial community composition changed over time.

Sp9509 was repeatedly surface sterilized with bleach to acquire gnotobiotic plants for this study. Despite taking measures to ensure Sp9509 sterility after our bleach treatments, such as plating duckweed onto solid LB media and checking for microbial growth at 28˚C for up to 3 days and conducting PCR using 16S-23S IGS primers to detect bacteria DNA, 16S rRNA amplicon sequencing of initial Sp9509 tissue (DAB t0) nevertheless captured 83 bacterial ASVs (S6 File). However, the number of bacterial ASVs found in the DAB t0 community was significantly lower (Wilcoxon rank sum test, p-value $< 0.05$) when compared to the WWDAB community (S4 Fig). This amount is similar to what was found in the roots of surface-sterilized rice seedlings [10]. Only a few of these DAB t0 ASVs (n = 11) were stable and contained a higher than average abundance (median clr $> 0$) in the WWDAB t10 community (S6 File). These stable ASVs were members of common plant-associated bacterial taxa such as *Burkholderia*, *Bacillus*, and *Pseudomonas* [5,7]. This may explain their strong association with Sp9509 and the difficulty encountered in sterilization of Sp9509. Surprisingly, from this analysis we observed that 884 bacterial ASVs were found only in the WWDAB community but not in the DAB t0 or WW t0 communities (S5 Fig). A majority of these ASVs (n = 769) were found in a small number of samples ($< 25\%$) and in low abundance (median clr $< 0$) (S5 Fig). These ASVs could either represent rare or random ASVs that were captured in the larger WWDAB sample size (n = 16) but not in the smaller WW t0 sample size (n = 2) or may represent technical artifacts generated by amplicon sequencing. The remaining ASVs (n = 115) were found in

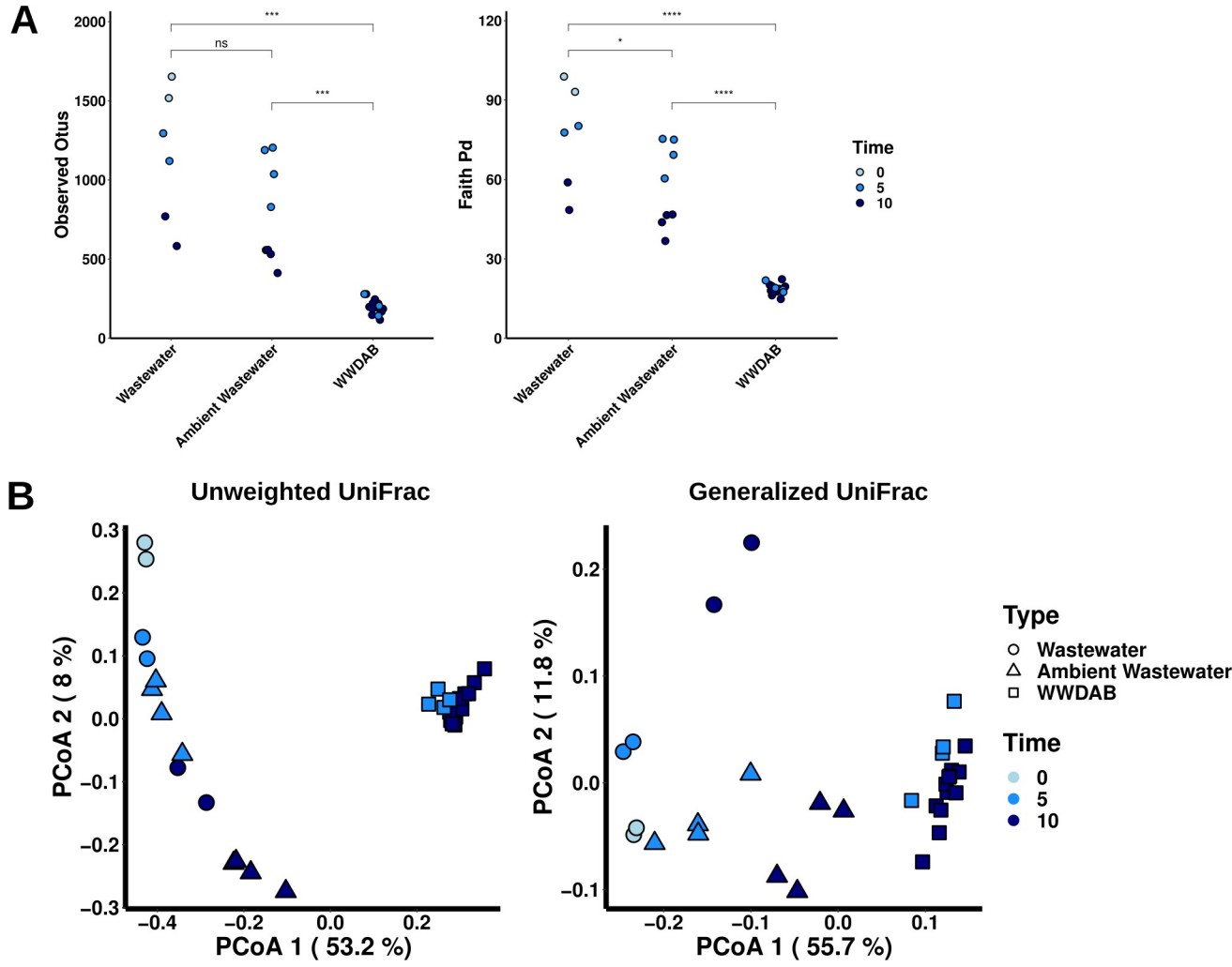

**Fig 4. A discrete bacterial community steadily assembles onto duckweed. (A)** The number of ASVs and Faith's PD index for wastewater (WW), ambient wastewater (AWW), and Sp9509 bacterial communities (WWDAB) derived from Princeton Meadows 2015 wastewater inoculum (Wilcoxon rank sum test; p-value < 0.05 = "*", p-value < 0.01 = "**", p-value < 0.001 = "***", p-value < 0.0001 = "****"). **(B)** Principal coordinate analyses using unweighted (left) and generalized (right) UniFrac distances between WW, AWW, and WWDAB communities.

several samples (> 25%) but only a smaller group (n = 19) had a greater than average abundance (median clr > 0) (S6 File). These 19 ASVs were all present in the AWW community and included members of common plant-associated families such as Bradyrhizobiaceae, Comamonadaceae, Oxalobacteraceae, and Sphingomonadaceae [7]. Therefore, these particular ASVs may have been too low in abundance in the wastewater to be detectable but were then enriched in AWW and DAB communities.

We rarefied samples (112500 reads) to examine bacterial community diversity and assembly (S2 Fig). The WWDAB community contained less ASVs and a lower Faith's PD index than the WW and AWW bacterial communities (Fig 4). PCoA results using the UUF distance revealed a profound and significant separation of the WWDAB community from the wastewater communities along the first principal coordinate (Fig 4 and S1 Table). Time contributed some variation to WW and AWW bacterial communities but did not have any observable influence on WWDAB community composition (Fig 4 and S1 Table). These findings show only a subset of the microbiota in the wastewater was assembled onto quasi-gnotobiotic

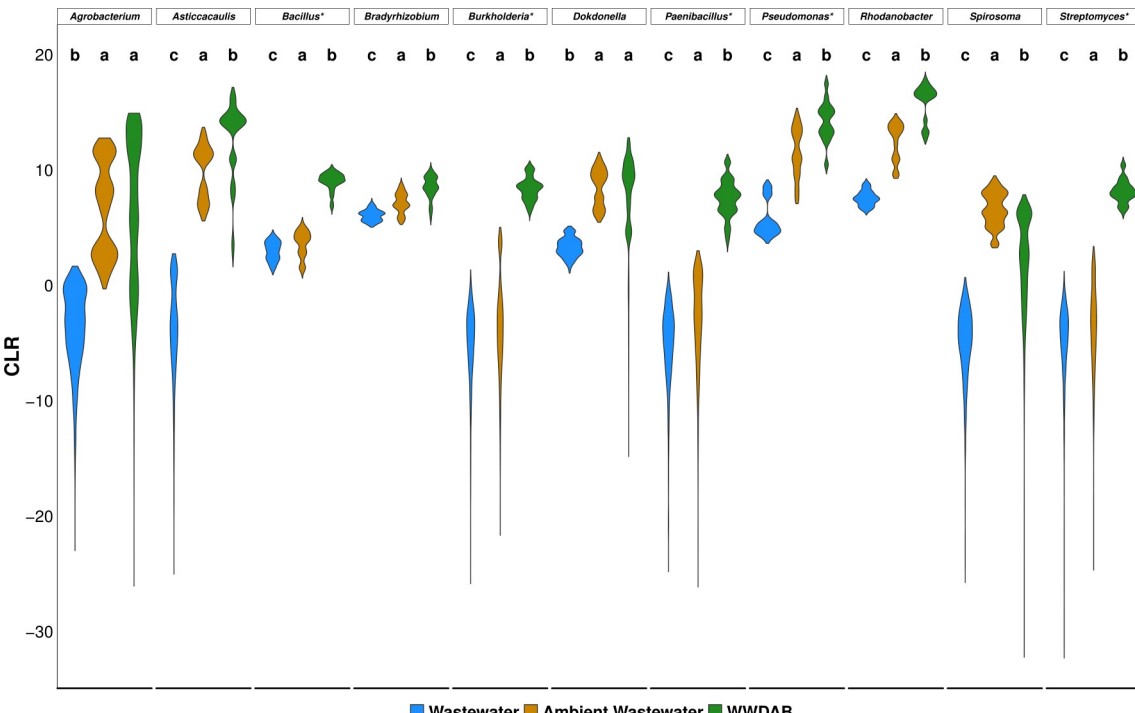

**Fig 5. Bacterial genera are enriched in both the ambient water and DAB community.** ALDEx2 analysis determined 12 significantly enriched bacterial taxa in the ambient water and WWDAB communities compared to the wastewater community (adjusted Welch's t-test, p-value < 0.05, absolute effect size greater than 1.50). The ALDEx2 distribution for each of these bacterial taxa are displayed. Multiple comparisons were performed using Dunn's test with command letters displayed. * = bacterial taxa for which bacterial ASVs were found in DAB t0 community, CLR = centered-log ratio.

Sp9509 similar to natural communities found on wild duckweed and their environmental microbiota (Fig 2). Once assembled, the respective DAB community remained stable over the time period we studied here.

We then analyzed the taxonomic structure of the bacterial communities and tested for differential abundance to ascertain community differences. Similar to the natural DAB community found in wild duckweed, Proteobacteria prevalently assembled onto duckweed (95% mean relative abundance) (S6 Fig). Family and genus level differences were observed between communities and the different time points, but these bacteria were only found in a few samples in low abundance (S6 Fig and S7 File). To verify these differences at the genus level, we tested for differential abundance. Pairwise comparison revealed 13 bacterial genera differed in abundance between WW and AWW communities, 39 genera differed between WW and WWDAB communities, and 25 genera differed between AWW and WWDAB communities (adjusted Welch's t-test, p < 0.05; absolute effect size > 1.5) (S8 File). Several Proteobacteria taxa significantly decreased in abundance in the WWDAB community compared to the water communities confirming that only certain Proteobacteria integrate into the WWDAB community (S8 File). Interestingly, bacteria enriched in the WWDAB community were also enriched in the AWW community compared to the WW community (Fig 5). *Bacillus*, *Burkholderia*, *Paenibacillus*, and *Streptomyces* had a significantly greater abundance in the WWDAB community compared to AWW community but many of these taxa were also found in the DAB t0 community (Fig 5 and S6 File). Therefore, it appears that microbiota members may be recruited into the surrounding water to facilitate incorporation into the duckweed microbiome similar to what has been observed in the rice microbiome [10]. Pairwise comparison between

communities at 5-day and 10-day post inoculation determined only *Rhodanobacter* increased and *Janthinobacterium* decreased in abundance over time within the WWDAB community demonstrating the duckweed microbiome was stable within the time periods tested (S9 File).

**Princeton Meadows year 2 study.** To determine the impact of host duckweed species on DAB community assembly and composition, we inoculated *Lemna minor* 5576 (Lm5576) and *Spirodela polyrhiza* 9509 (Sp9509) with Princeton Meadows wastewater effluent collected in the summer of 2016 (S3 File). Samples were collected at 0, 2, and 7 days post inoculation. Plant tissues are commonly treated to compartmentalize the plant microbiome [8]. To observe the effect of tissue pre-treatment on the duckweed microbiome, we treated Lm5576 and Sp9509 either with water to wash off any loosely bound bacteria or with salt and detergent (SD) solution to remove attached epiphytes. Those bacteria remaining after SD treatment are assumed to represent strongly attached epiphytes and/or endophytes. Similar to the Princeton Meadows year 1 study, initial quasi-gnotobiotic duckweed tissues (DAB t0) contained a total of 307 bacterial ASVs but their number of reads were significantly lower when compared to wastewater-inoculated duckweed (WWDAB) (S4 Fig). Moreover, only 40 of these ASVs contained a higher than average abundance at 7 days after wastewater inoculation (S10 File). Most of these stable bacterial ASVs came from known plant-associated Proteobacteria families such as Comamonadaceae, Oxalobacteraceae, and Pseudomonadaceae suggesting a conserved role for their interaction with duckweed [5,7].

Differential abundance testing and between sample diversity analysis were implemented to test the impact of tissue treatment, time, and host duckweed species on DAB community composition. Tissue treatment did not significantly alter DAB community structure while time resulted in significant change of DAB community composition (S11 File and S1 Table). A generalized linear model and pairwise comparison revealed a number of bacterial taxa (5 and 12 respectively) were significantly altered in abundance between t2 and t7 communities (S11 File). Additionally, between sample diversity analysis revealed significant variation in DAB communities at the different time points (Fig 6 and S1 Table). Time had a greater influence on DAB community structure in this study compared to the Princeton Meadows year 1 study and this might be because communities were analyzed at earlier time points. This suggests that some minimum incubation time (t > 2 day) may be required for the DAB community to stabilize. Differential abundance testing concluded only *Xanthobacter* differed in abundance between Lm5576 (LmDAB) and Sp9509 (SpDAB) bacterial communities (S11 File) while diversity analysis did not demonstrate significant variation between LmDAB and SpDAB communities (Fig 6 and S1 Table). Most taxa in both LmDAB and SpDAB communities were found in similar abundance (Fig 6). Together, these data indicate similar bacterial communities asssociate with different duckweed species.

## Comparative analysis of bacterial communities

Core plant microbiota represent a subset of microbes that are consistently found in the plant microbiome [8,9]. Here, we conducted a cross-study comparison of DAB consortium to identify core members in the duckweed microbiome. DAB communities collected from the Caldwell House, Passion Puddle, Princeton Meadows year 1 and Princeton Meadows year 2 studies along with DAB communities from an ecological study comparing rice tissues and duckweed microbiomes in 3 Chinese rice paddies [38] were included (S1 and S3 Files). To identify this group of microbiota, we first distinguished a set of 24 "core" bacterial taxa that were present in at least 6 communities since the Caldwell House DAB community harbored less taxa than the other communities (Fig 7 and S7 Fig). Next, we compared the abundance of each core taxa to the remaining non-core community. Bacterial communities can be composed of many taxa

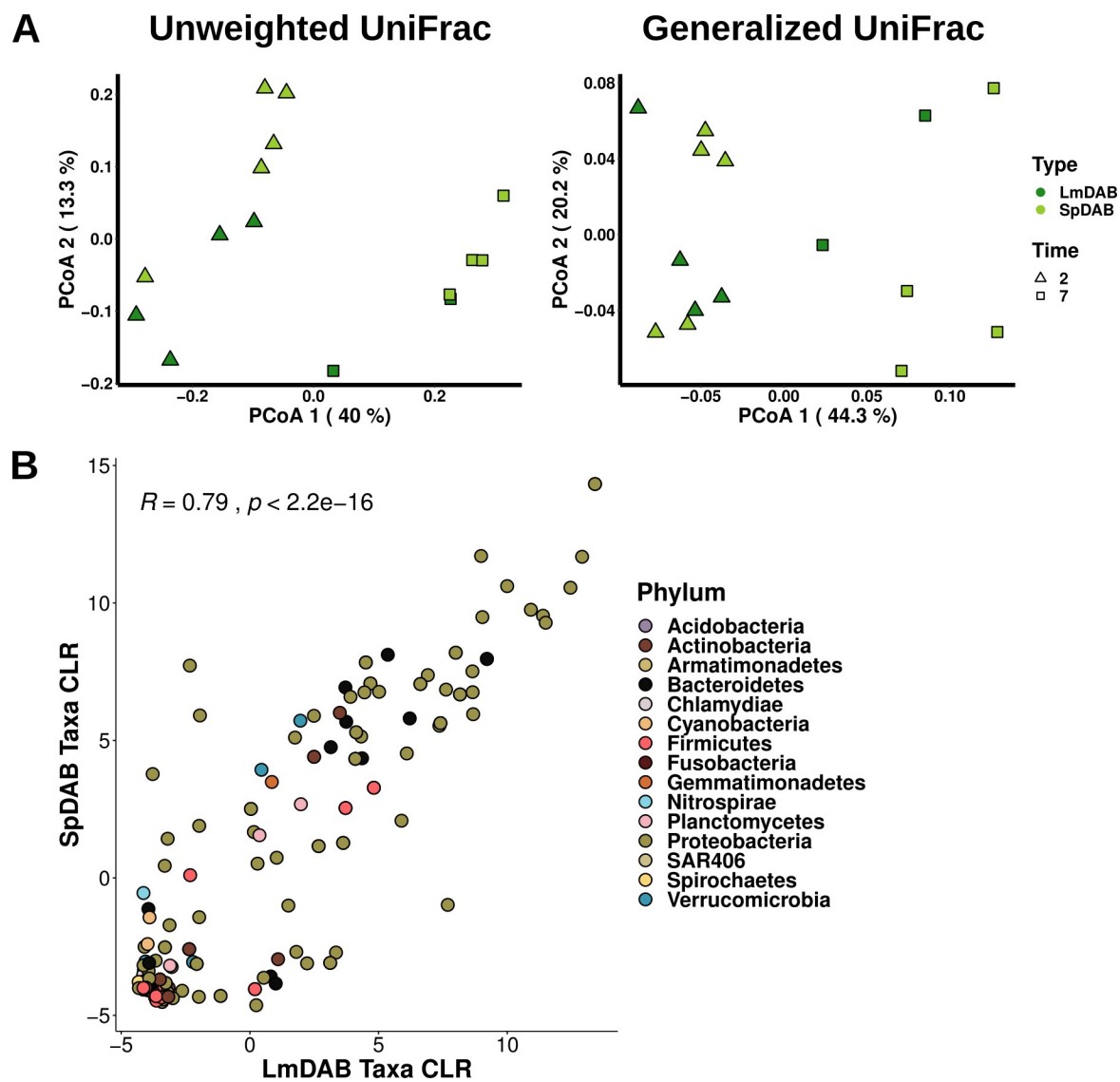

**Fig 6. Different duckweed species host similar bacterial communities. (A)** Principal coordinate analysis of Sp9509 (SpDAB) and Lm5576 (LmDAB) DAB communities using unweighted (left) and generalized (right) UniFrac distances. **(B)** Scatterplot of bacterial taxa abundance between SpDAB and LmDAB bacterial communities assembled from Princeton Meadows year 2 inoculum. Pairwise Spearman rank correlation coefficient and p-value are displayed. CLR = median centered-log ratio for taxa.

but most taxa are in low abundance [41]. Therefore, we re-defined core members as micro-biota with a 2-fold greater abundance than the non-core community in at least 6 studies. With this criteria, 11 Proteobacteria taxa constituted the duckweed core microbiome (Fig 7). Most members of the DAB core microbiome such as *Acinetobacter*, *Agrobacterium*, *Azospirillum*, *Burkholderia*, *Caulobacter*, *Methylibium*, *Pseudomonas*, and *Sphingomonas* are also found in the terrestrial plant microbiome and can play pivotal roles in plant health and growth promotion [11,12,42,43].

We then compared the bacterial taxa composition of duckweed-, rice-, and Arabidopsis-associated bacterial communities to discern differences between aquatic and terrestrial plant microbiomes. Since the Lemnaceae family diverged from the major monocot lineage, which includes rice, of angiosperms more than 100 Mya when it returned to a completely aquatic

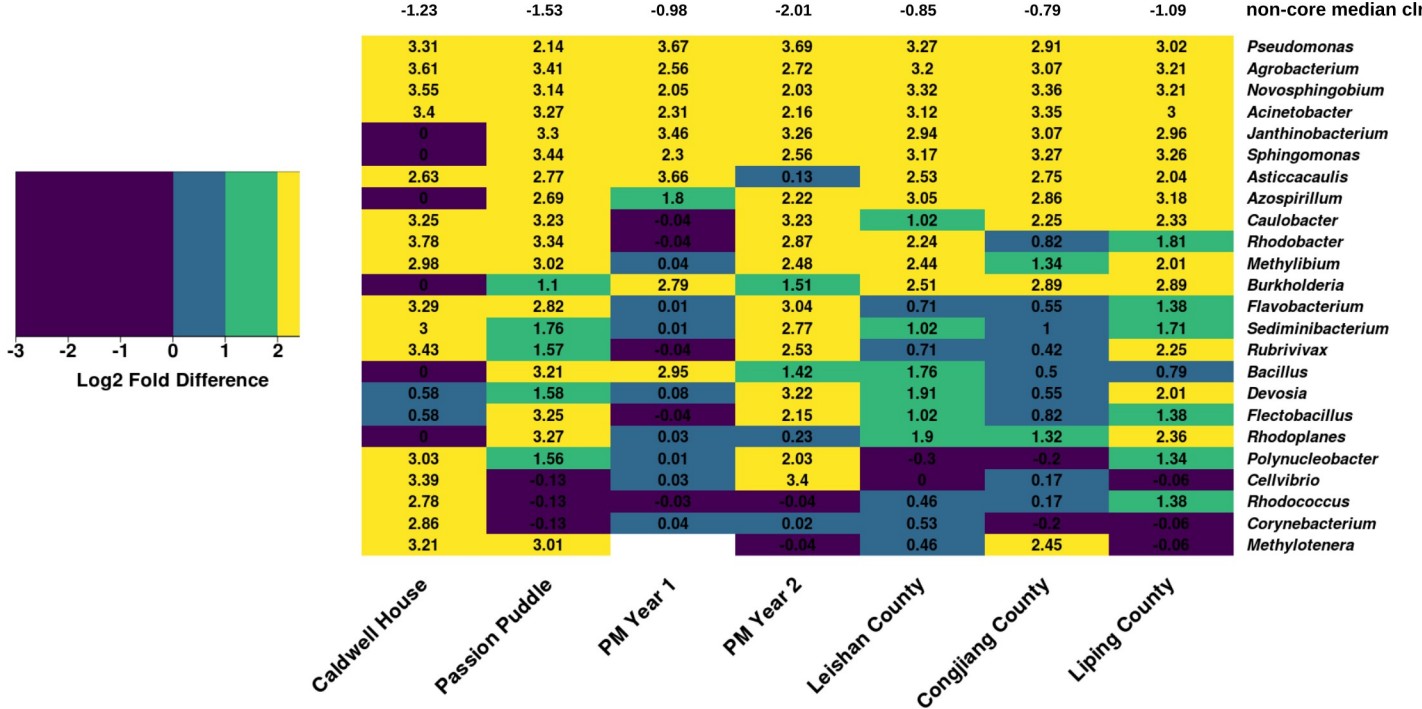

**Fig 7. Bacterial taxa in the duckweed core microbiome.** Core taxa were found in at least 6 of the 7 DAB communities analyzed. Displayed is the log2 fold difference between core taxa median centered-log ratio (clr) to non-core community median clr from Caldwell House, Passion Puddle, Princeton Meadows years 1–2, Leishan County (China), Congjiang County (China), and Liping County (China) DAB community studies. Taxa were considered core members if they displayed a 2-fold (log2 > 1) higher abundance in at least 6 studies. Negative values signify taxa abundance was lower than non-core community abundance. Abundance was found to be significantly different (p-value < 0.05) between all core taxa and non-core community comparisons using Dunnett's test with the non-core community as a control. The median non-core community clr is displayed for each of the 7 DAB communities. No *Methylotenera* taxa were found in DAB communities from the Princeton Meadows year 1 study.

habitat [23], we would expect that it may have evolved novel associations with microbial partners in this type of environment while retaining other conserved ones. For this comparison, the DAB communities mentioned above were used to assemble the duckweed microbiome while two microbiome studies from rice [10,38] and two Arabidopsis microbiome studies [36,37] were used to construct the terrestrial plant microbiome (S1 File). A few of these studies divided the plant microbiome into different compartments. To allow for direct comparison to the duckweed microbiome, we combined these different plant compartments into one representing the Arabidopsis- or rice-associated bacterial community (S2 File). Between community diversity was calculated using the Jaccard distance to assess community composition and the Bray-Curtis distance to assess community abundance (S1 Table). PCoA using the Bray Curtis distance showed the separation of the respective plant-associated bacterial communities into three groups encompassing: 1) the leaf-associated bacterial community of all three hosts, 2) the rice root-associated bacterial community, and 3) the Arabidopsis root-associated bacterial community (Fig 8). Interestingly, the identity of the plant host did not contribute any significant variation to community structure (S1 Table). Actinobacteria, Bacteroidetes, Firmicutes, and Proteobacteria are recognized as the predominant phyla in the plant-associated bacterial microbiome [5]. Analysis of the quantity of bacterial taxa from each of these phyla showed significantly fewer taxa from Actinobacteria on monocot leaves compared to dicot leaves and terrestrial plant root samples (Fig 8). These data suggest a highly conserved structuring effect of leaf tissue on the plant bacterial microbiome with monocot leaves hosting less Actinobacteria.

## Discussion

### Factors driving assembly of the duckweed-associated bacterial community

Culture-independent characterization has elucidated some of the key determinants involved in root microbiome assembly and composition in model plant species. Many of these studies

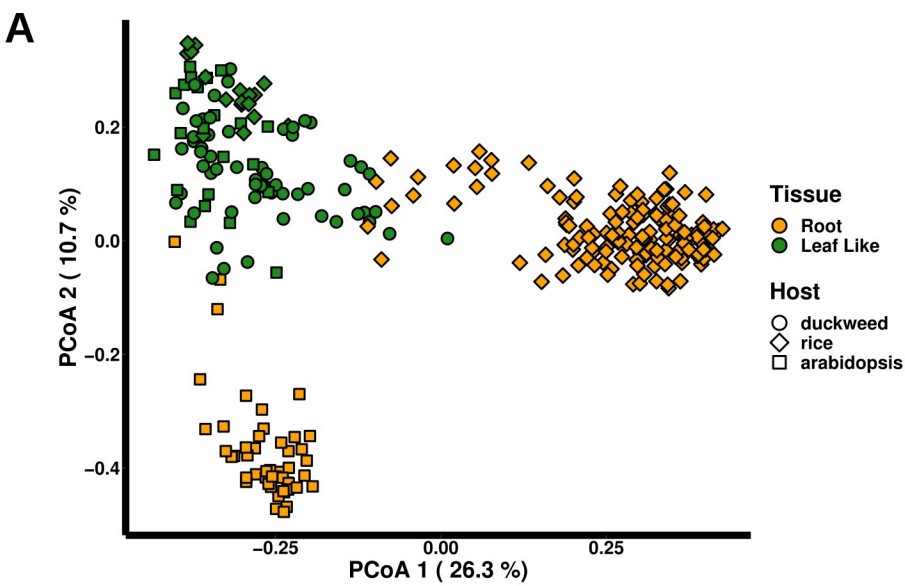

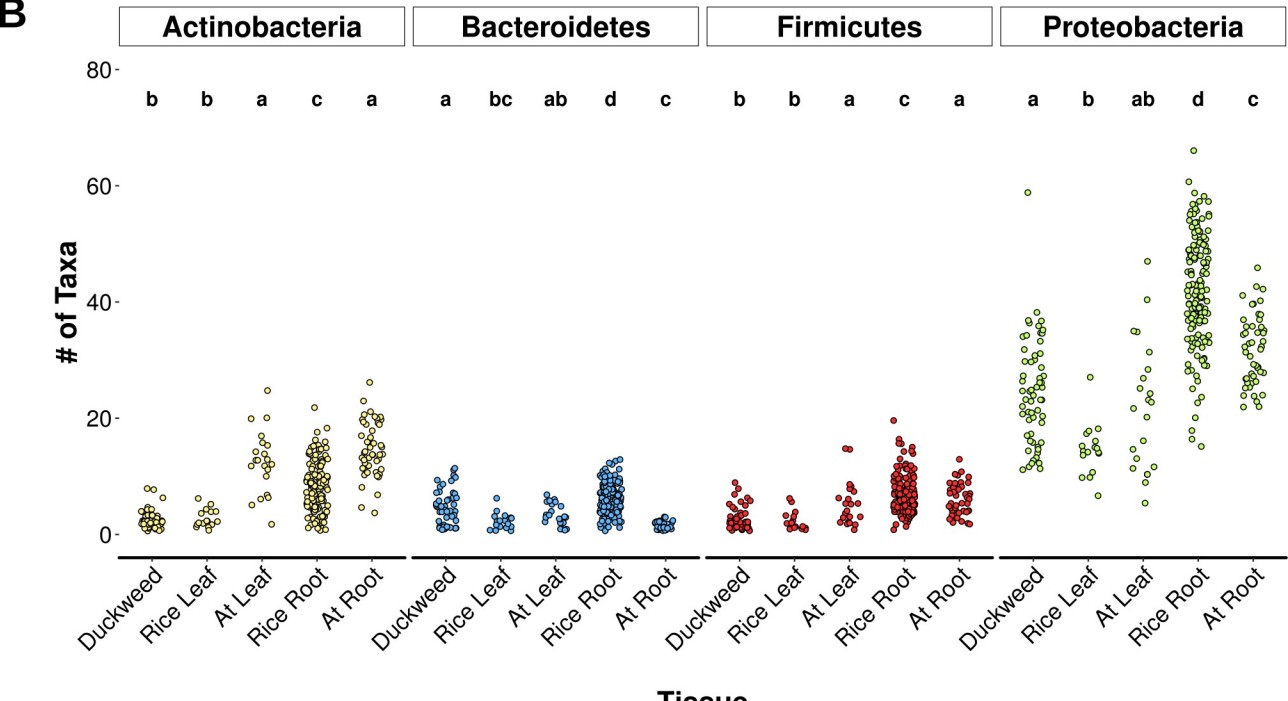

**Fig 8. The duckweed bacterial microbiome resembles the terrestrial leaf microbiome.** The bacterial taxa composition of duckweed, rice, and Arabidopsis bacterial communities were compared. Samples were rarefied to 1000 reads. **(A)** PCoA using the Bray-Curtis distance for duckweed, rice, and Arabidopsis bacterial communities. **(B)** The number of bacterial genera from four predominant plant-associated bacterial phyla was calculated for different plant tissues. Multiple pairwise comparison testing was performed using Dunn's test with Benjamini-Hochberg adjustment and the resultant compact letters are displayed.

show distinct communities formed on plants compared to the surrounding soil [7–10]. In particular, one landmark study of the rice root microbiome found that microbial diversity decreased along the soil, rhizosphere, rhizoplane, and endosphere nexus [10]. Additionally, they observed a majority of bacteria enriched in the rhizoplane/endosphere were also enriched in the rhizosphere. Therefore, they proposed a multi-step acquisition model for the root microbiome where select taxa are enriched in the rhizosphere and only some of these taxa are able to colonize the rhizoplane and endosphere. Here we used the community profiling approach, along with microbial ecology methods and differential abundance testing, to investigate assembly of the duckweed bacterial microbiome. Duckweed, as a floating aquatic plant, provides a unique opportunity to investigate assembly of the phyllosphere since its leaf tissues (fronds) are in direct contact with the inoculum (ambient water), similar to roots in soil. Bacterial communities from both wild duckweed and quasi-gnotobiotic duckweed inoculated with wastewater effluent showed a distinct phylogenetic signature compared to the surrounding water (Figs 2 and 4). Moreover, comparative analysis across several different DAB communities identified a set of core bacterial microbiota abundantly associated with duckweed across a large range of samples from different locales and conditions (Fig 7). In addition, a similar recruitment dynamic like that observed in the rice root microbiome was observed for the DAB community. In our Princeton Meadows year 1 study we found diversity decreased along the wastewater, ambient wastewater, and duckweed nexus (Fig 4 and S8 File). While many bacteria decreased in abundance along this nexus, bacteria enriched on duckweed were also enriched in ambient wastewater when compared to wastewater without duckweed (Fig 5). Soil site or origin is the major determinant of the root bacterial community even at a continental scale [7–10,44]. We found the duckweed bacterial microbiome was remarkably constant across the locations examined, demonstrating a similar phylogenetic profile (Fig 2). While taxonomic analysis showed a few bacterial taxa differed between locations, they were found not to be significant (Fig 3 and S5 File). Plant host species is a minor determinant of the root microbiome with communities from highly diverged species showing only quantitative differences [9,10]. Here, the duckweed microbiome was not affected by duckweed host species. *L. minor* and *S. polyrhiza* inoculated with the same wastewater effluent hosted similar communities as revealed by diversity analyses, differential abundance testing, and cross-comparison of bacterial community member abundance (Fig 6). Time series experiments in the rice root microbiome demonstrated time had a significant effect on community composition [10]. It was determined that microbiome acquisition can start as early as 24 hours while microbiome stability may take as long as 2 weeks to achieve. The duckweed microbiome is stable once it is assembled, as illustrated by 5-day and 10-day DAB communities in the Princeton Meadows year 1 study, but may take some time to stabilize, as suggested by the Princeton year 2 study between 2-day and 7-day DAB communities (Fig 6 and S11 File). The inoculation studies presented here suggest that microbiome stability is reached as early as 5 days in duckweed compared to 2 weeks in the rice root microbiome [10]. One possible scenario may be that: first, duckweed acquires its microbiome faster because of its aquatic habitat, where microbes may easily navigate through solution compared to soil in terrestrial environments. Secondly, unlike other plants, duckweed reproduces axsexually where a daughter frond emerges from a mother frond [13]. Only fronds are produced throughout this developmental cycle, in contrast to terrestrial plants where new and more complex structures may arise throughout development changing microbiota dynamics. Therefore, once plant microbiota colonize, exposure to the same duckweed frond tissue over time may allow community interactions to stabilize much quicker. Together, these data suggest conserved structuring principles between duckweed and terrestrial plant microbial communities.

## Structure of the duckweed-associated bacterial microbiome

Proteobacteria is one of the major phyla found in the plant microbiome [5,7]. Genomic analysis of bacterial microbiota genomes from Arabidopsis showed Proteobacteria contained the highest functional diversity compared to other plant-associated bacterial phyla [11]. They found Proteobacteria formed distinct functional clusters based on family taxonomy rather than by ecological niche. Further investigation identified carbohydrate and xenobiotic degradation as enriched gene categories in plant-associated microbiota [11,42]. In line with proposed plant microbiome acquisition models these enriched gene categories may allow plant microbiota to establish a presence in the surrounding environment prior to their association with the respective plant host [10,11]. Bacterial community profiles of duckweed collected from natural sites and inoculated with wastewater effluent revealed Proteobacteria as the prevailing phylum in the duckweed bacterial microbiome (Fig 3 and S6 Fig). Different Proteobacteria taxa may be enriched in different scenarios but some taxa are conserved throughout (Fig 7 and S3 Fig). *Pseudomonas* and *Acinetobacter* were two prominent Proteobacteria genera found in the "core" duckweed microbiome (Fig 7). Recent investigations involving synthetic communities revealed *Pseudomonas* as one of the key bacterial taxa involved in plant-microbiota mediated immunity against filamentous eukaryotic microbes [12,45]. Duckweed inoculated with an *Acinetobacter* strain promoted duckweed growth under sterile and non-sterile conditions, protected against indigenous microbes, and increased bioremediation capability through the degradation of phenol [46,47]. Interestingly, mono-association studies typically show only a transient colonization of host duckweed plants by a particular bacteria strain suggesting that ecological interactions among different members of a plant microbiome may be critical for long term stability of the microbial community [2,3,37,48]. Moreover, despite using both culture-dependent and molecular measures to ensure complete duckweed sterility for our studies, we observed some bacterial reads in our quasi-gnotobiotic duckweed samples. The omnipresence of bacteria on duckweed and the resultant contamination is an issue encountered in many different projects across many different laboratories working with duckweed. Interestingly, many of the bacterial reads in our quasi-gnotobiotic duckweed turned out to belong to the Proteobacteria core taxa (S6 and S10 Files, Fig 7). These common contamination issues and the prevalence of Proteobacteria in the duckweed microbiome indicate a strong interaction between duckweed and Proteobacteria while results from functional experiments implicate fundamental roles of select bacterial taxa from this phylum in plant protection and growth promotion. In addition, PGPB stability or persistence is an important aspect of successful PGPB application [2]. Many studies show that exogenous PGPB disappear from the resident microbial community within a few weeks after application [2]. Therefore, one possible strategy to improve PGPB performance could therefore be to select PGPB from stable host "core" microbiota.

Some compositional differences can exist between plant leaves and roots, with leaves hosting a greater abundance of Proteobacteria than roots [6]. Despite considerable taxonomic and functional overlap between the root and leaf microbiota, certain community members are better able to colonize their original plant organ [11]. Here we compared duckweed and terrestrial leaf and root bacterial communities. Even though whole duckweed plants were used in this study, fronds compose the majority of duckweed biomass. The duckweed bacterial community matched the terrestrial leaf microbiome of rice and Arabidopsis while it was clearly distinguished from the root microbiomes of terrestrial plant hosts (Fig 8). There were significantly fewer Actinobacteria taxa in monocot leaf communities compared to the terrestrial plant root bacterial community (Fig 8). Interestingly, Actinobacteria encompasses a distinct clade of bacteria referred to as the terrabacteria [49]. Terrabacteria evolved from a common ancestor on

land and acquired traits such as a peptidoglycan layer and spore formation to help withstand stresses commonly found in terrestrial environments such as UV radiation, high salt concentrations, and drought [49]. Several investigations into the effect of drought on the plant root microbiome showed an enrichment of Actinobacteria under drought conditions [50–53]. Furthermore, investigation into the assembly cues of the Arabidopsis microbiome showed Actinobacteria were specifically enriched in Arabidopsis roots and required additional host cues while other phyla, such as Proteobacteria, colonized inactive lignocellulosic matrices suggesting general plant-cell wall features were sufficient colonization cues [54]. Therefore, we hypothesize that terrestrial plants may recruit Actinobacteria species into their root-associated bacterial communities to facilitate adaptation to stresses commonly encountered in terrestrial environments, such as drought, while aquatic duckweed may not have such needs. Moreover, the similarity found between duckweed and terrestrial leaf microbiomes suggests a conserved organizational influence of plant leaf tissue over a large evolutionary distance of 100 Myr.

In conclusion, this report demonstrates the utility of using duckweed to study the plant microbiome. Results from our survey of wild duckweed tissues and inoculation studies showed duckweed exhibits bacterial community structuring principles similar to those of terrestrial plants. Analyses of taxa composition revealed a similar taxonomic structure between the duckweed bacterial microbiome and terrestrial leaf microbiome, with less Actinobacteria in the DAB community. These data suggest a conserved structuring effect by leaf tissue on plant microbiota. Furthermore, we present a set of duckweed core microbiota that can be selected and further studied for stable PGPB behavior in this aquatic model plant system.

## Supporting information

**S1 Fig. QIIME 2 feature-classifier classified a majority of reads at the genus level.** Different methods for the q2 feature-classifier plugin were tested for their ability to classify ASVs at the genus level. The effect of using different databases was tested by either using the Greengenes 13_8 99% OTUs reference database (gg) or SILVA 132 (silva) database. Default parameters were used in each instance. dada2 = reads remaining after q2-dada2 quality control, NB = q2 feature classifier using naive bayes method, blast = q2 feature classifier using BLAST+ consensus method, vsearch = q2 feature-classifier using VSEARCH consensus method.
(TIF)

**S2 Fig. Rarefaction of bacterial communities generated in this study. (A)** The number of ASVs observed in Caldwell House and Passion Puddle bacterial communities at different sampling depths. The solid line intercepting the x-axis represents a sampling depth of 3664 reads. **(B)** The number of ASVs observed in Princeton Meadows year 1 bacterial communities at different sampling depths. Samples were rarefied to 112500 reads. **(C)** The number of ASVs observed in Princeton Meadows year 2 bacterial communities at different sampling depths. Samples were rarefied to 108000 reads.
(TIF)

**S3 Fig. Taxa enriched in duckweed bacterial communities compared to ambient water are different between Caldwell House and Passion Puddle sites.** ALDEx2 was performed to determine bacterial genera whose abundance was significantly different between duckweed-associated bacterial (DAB) community and ambient water communities from Caldwell House and Passion Puddle. Violin plots display the distribution of centered-log ratios (CLR) for bacterial genera whose abundance was significantly different between communities ("*" = adjusted Welch's t-test, p-value < 0.05). Effect sizes are displayed for each bacterial genus. Larger values signify a greater difference between communities. Positive effect sizes represent

a higher abundance in DAB community compared to ambient water community while negative effect sizes represent a higher abundance in ambient water community compared to DAB community.
(TIF)

**S4 Fig. Comparison of bacterial reads found in quasi-gnotobiotic duckweed versus wastewater-inoculated duckweed.** Number of plastid and bacteria reads normalized to total reads in initial quasi-gnotobiotic Sp9509 (DAB t0) compared to Sp9509 duckweed tissue several days after inoculation with wastewater from both Princeton Meadows year 1 and year 2 studies (WWDAB). Pairwise comparison was performed using Wilcoxon rank sum test with p-values displayed.
(TIF)

**S5 Fig. Analysis of bacterial ASVs from the Princeton Meadows Year 1 study. (A)** Venn diagram showing the number of bacterial ASVs specific to and shared between the initial Sp9509 tissue (DAB _t0), initial wastewater inoculum (WW_t0), and Sp9509 inoculated with wastewater (WWDAB). **(B)** For each bacterial ASV found in the WWDAB community (n = 16), we calculated abundance (median clr), the amount of samples the ASV was found in (ASV Prevalence Category), and the communities the ASV was found in (WWDAB, WW_t0, DAB_t0). Each data point in the graph represents a bacterial ASV.
(TIF)

**S6 Fig. Proteobacteria predominantly assemble into the DAB community. (A)** Phylum, and **(B)** family level composition of Princeton Meadows year 1 bacterial communities. **(C)** Number of bacterial genera specific to and shared between bacterial communities and time points.
(TIF)

**S7 Fig. Percentage of samples each bacterial taxa was observed in.** DAB communities from different studies were analyzed for the presence of bacterial taxa from the phyla Actinobacteria, Bacteroidetes, Firmicutes, and Proteobacteria. The percentage of samples each bacterial taxa was observed in for each location was calculated. The color black illustrated in heatmaps means bacterial taxa was not observed in any samples for that location. Actinobacteria, Bacteroidetes, and Firmicutes genera that were observed in more than 1 study are displayed while Proteobacteria taxa in more than 4 studies are displayed.
(TIF)

**S1 File. Sample metadata and library information for rice and Arabidopsis microbiome studies used to construct the terrestrial plant microbiome.** Metadata were compiled from references [10,36–38] and libraries processed using QIIME 2.
(XLSX)

**S2 File. Paired-sample metadata files used to decompartmentalize rice and Arabidopsis microbiome studies.** Epiphytic and endophytic sample fractions from references [10,36–38] were paired.
(XLSX)

**S3 File. Sample metadata and library Information.** Excel spreadsheet containing metadata and library processing stats for samples from Caldwell House and Passion Puddle, Princeton Meadows year 1, and Princeton Meadows year 2 studies.
(XLSX)

**S4 File. Taxa information for Caldwell House and Passion Puddle bacterial communities.** Phylum and family level relative abundance of Caldwell House and Passion Puddle bacterial

communities. Information on all bacteria associated with duckweed from Caldwell House and Passion Puddle, bacteria specific to Caldwell House duckweed and specific to Passion Puddle duckweed, bacteria found in Caldwell House duckweed but not ambient water, and bacteria found in Passion Puddle duckweed but not ambient water. RA = relative abundance, Sample_Percent = percentage of samples genus was found in, OTU_Count = number of ASVs within bacterial genus, Mean_RA_Rank = bacterial genera were ranked by their mean relative abundance, sample_type = indicates whether genus was observed in ambient water (AW) and/ or DAB (DAB) community.
(XLSX)

**S5 File. Differential abundance testing of Caldwell House and Passion Puddle bacterial communities using ALDEx2.** Differential abundance testing between ambient water (AW) and DAB communities from Caldwell House and Passion Puddle. Differential abundance testing between Caldwell House (CH) and Passion Puddle (PP) ambient water and DAB communities. rab.win.ambient_water = median clr value in ambient water community, rab.win. rinsed_tissue = median clr in DAB community, diff.btw = median difference in clr values between DAB and ambient water communities, diff.win = median of largest difference in clr values within DAB and ambient water communities, effect = median effect size (diff.btw / diff. win), we.eBH = expected benjamini-hochberg corrected p-value of Welch's t-test, wi.eBH = expected benjamini-hochberg corrected p-value of Wilcoxon rank test, rab.win.Caldwell_ House = median clr value in Caldwell House community, rab.win.Passion_Puddle = median clr value in Passion Puddle community.
(XLSX)

**S6 File. Bacterial ASVs found in duckweed bacterial communities from the Princeton Meadows Year 1 study.** This file includes the number of reads for each bacterial ASV found within initial Sp9509 tissue (DAB_t0), the median centered-log ratio of the 35 ASVs found in both DAB_t0 and wastewater-inoculated Sp9509 (WWDAB) communities at different time points, and information for each bacterial ASV found in the WWDAB community including abundance (median clr), percentage of samples ASV was found in (ASV Prevalence Category), and communities ASV was observed in (WW_t0, DAB_t0, WWDAB). WW_t0 = initial wastewater inoculum.
(XLSX)

**S7 File. Taxa information for Princeton Meadows year 1 bacterial communities.** Phylum and family level relative abundance for Princeton Meadows year 1 bacterial communities and information for bacterial taxa specific to WWDAB community, WWDAB t5 specific taxa, and WWDAB t10 specific taxa. RA = relative abundance, Sample_Percent = percentage of samples bacterial genus was found in, OTU_Count = number of ASVs from bacterial genus, Mean_RA_Rank = rank was assigned according to mean relative abundance.
(XLSX)

**S8 File. Differential abundance testing between different bacterial communities from Princeton Meadows year 1 study using ALDEx2.** Differential abundance testing of bacterial genera between wastewater (WW), ambient wastewater (AWW), and Sp9509-associated bacterial (WWDAB) communities using Kruskal-Wallis test and a generalized linear model. Also includes pairwise comparisons between WW and AWW communities (taxa with effect sizes less than -1 are more abundant in AWW than WW), WW and WWDAB communities (taxa enriched in WWDAB community have effect sizes less than -1), and WWDAB and AWW communities (taxa enriched in WWDAB community have effect sizes greater than 1). rab.win. ambient_water = median clr value in AWW community, rab.win.wastewater = median clr

value in WW community, rab.win.treated_tissue = median clr in WWDAB community, we.
eBH = expected benjamini-hochberg correct p-value of Welch's t-test, diff.btw = median difference in clr values between communities, diff.win = median of largest difference in clr values within community, effect = median effect size (diff.btw / diff.win).
(XLSX)

**S9 File. Differential abundance testing of Princeton Meadows year 1 bacterial communities between different time points using ALDEx2.** Pairwise comparison between 5 day and 10 day ambient wastewater (AWW) and Sp9509-associated bacterial (WWDAB) communities. rab.win.5 = median clr in t5 community, rab.win.10 = median.clr in t10 community.
(XLSX)

**S10 File. Analysis of bacterial ASVs found in initial duckweed tissue used for the Princeton Meadows Year 2 study.** This file contains the number of reads for each bacterial ASV found in initial quasi-gnotobiotic duckweed (DAB t0) along with median centered-log ratio of bacterial ASVs found in both DAB t0 and wastewater-inoculated duckweed communities across different time points for both LmDAB and SpDAB communities.
(XLSX)

**S11 File. Differential abundance testing between Princeton Meadows year 2 bacterial communities.** ALDEx2 was used to generate a generalized linear model to determine if DAB community composition changed between host duckweed species, time points, and tissue treatments. In addition, pairwise comparisons of bacterial taxa abundance were made between LmDAB and SpDAB communities, t2 and t7 communities, and water and SD treated DAB communities. DAB t0 communities were excluded from testing. rab.win.L_minor370_DWC112 = median clr value in LmDAB communities, rab.win.S_polyrhiza432_9509 = median clr value in SpDAB communities, rab.win.2 = median clr value in t2 day DAB communities, rab.win.7 = median clr value in t7 DAB communities, rab.win.water_treated = median clr value in water treated DAB communities, rab.win.SD_treated = median clr value in salt and detergent treated DAB communities, we.eBH = expected benjamini-hochberg correct p-value of Welch's t-test, diff.btw = median difference in clr values between factors tested, diff.win = median of largest difference in clr values within factors tested, effect = median effect size (diff.btw / diff.win).
(XLSX)

**S1 Table. PERMANOVA results using unweighted (UUF) and generalized (GUF) unifrac distance metrics.**
(XLSX)

## Acknowledgments

Duckweed research at the Lam Lab is supported in part by a grant from the Department of Energy (DE-SC0018244) and a Hatch project (#12116) from the New Jersey Agricultural Experiment Station at Rutgers University. The Lebeis Lab acknowledges start-up funds provided to her from the University of Tennessee, Knoxville. Contribution by the Facilities Integrating Collaborations for User Science (FICUS) initiative and under Contract Nos. DE-AC02-05CH11231 (JGI) and DE-AC05-76RL01830 (EMSL) to the sequencing of the duckweed microbiome work is also gratefully acknowledged.

## Author Contributions

**Conceptualization:** Sarah Lebeis, Eric Lam.

**Data curation:** Kenneth Acosta.

**Formal analysis:** Kenneth Acosta.

**Funding acquisition:** Sarah Lebeis, Eric Lam.

**Investigation:** Kenneth Acosta, Jenny Xu, Sarah Gilbert, Elizabeth Denison, Thomas Brinkman.

**Methodology:** Kenneth Acosta, Sarah Lebeis, Eric Lam.

**Project administration:** Eric Lam.

**Resources:** Sarah Lebeis, Eric Lam.

**Software:** Kenneth Acosta.

**Supervision:** Sarah Lebeis, Eric Lam.

**Visualization:** Kenneth Acosta.

**Writing – original draft:** Kenneth Acosta.

**Writing – review & editing:** Kenneth Acosta, Sarah Lebeis, Eric Lam.

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
