## [Decision Letter · Decision Letter 0]

12 Nov 2019

PONE-D-19-25118

Duckweed hosts a taxonomically similar bacterial assemblage as the terrestrial leaf microbiome

PLOS ONE

Dear Dr. Lam,

Thank you for submitting your manuscript to PLOS ONE. After careful consideration, we feel that it has merit but does not fully meet PLOS ONE’s publication criteria as it currently stands. Therefore, we invite you to submit a revised version of the manuscript that addresses the points raised during the review process.

The two reviewers and myself agree that, fundamentally, the work reported is worthy of publication in PLoS ONE.  However, we also agree that a substantial rewriting of the manuscript is necessary for the work to be reader-friendly when it appears. The reviewers point out many of the difficulties in the already submitted manuscript. I will be glad to see a thorough revision.

We would appreciate receiving your revised manuscript by Dec 27 2019 11:59PM. To enhance the reproducibility of your results, we recommend that if applicable you deposit your laboratory protocols in protocols.io, where a protocol can be assigned its own identifier (DOI) such that it can be cited independently in the future. For instructions see: http://journals.plos.org/plosone/s/submission-guidelines#loc-laboratory-protocols

We look forward to receiving your revised manuscript.

Kind regards,

Ulrich Melcher

Academic Editor

PLOS ONE

Journal Requirements:

3. We note that Figure 1 in your submission contain map/satellite images which may be copyrighted. All PLOS content is published under the Creative Commons Attribution License (CC BY 4.0), which means that the manuscript, images, and Supporting Information files will be freely available online, and any third party is permitted to access, download, copy, distribute, and use these materials in any way, even commercially, with proper attribution. For these reasons, we cannot publish previously copyrighted maps or satellite images created using proprietary data, such as Google software (Google Maps, Street View, and Earth). For more information, see our copyright guidelines: http://journals.plos.org/plosone/s/licenses-and-copyright.

a).    You may seek permission from the original copyright holder of Figure(s) [#] to publish the content specifically under the CC BY 4.0 license.

b).    If you are unable to obtain permission from the original copyright holder to publish these figures under the CC BY 4.0 license or if the copyright holder’s requirements are incompatible with the CC BY 4.0 license, please either i) remove the figure or ii) supply a replacement figure that complies with the CC BY 4.0 license. Please check copyright information on all replacement figures and update the figure caption with source information. If applicable, please specify in the figure caption text when a figure is similar but not identical to the original image and is therefore for illustrative purposes only.

Reviewers' comments:

Reviewer's Responses to Questions

**Comments to the Author**

1. Is the manuscript technically sound, and do the data support the conclusions?

Reviewer #1: Yes

Reviewer #2: Yes

2. Has the statistical analysis been performed appropriately and rigorously? 

Reviewer #1: Yes

Reviewer #2: Yes

3. Have the authors made all data underlying the findings in their manuscript fully available?

Reviewer #1: Yes

Reviewer #2: Yes

4. Is the manuscript presented in an intelligible fashion and written in standard English?

Reviewer #1: Yes

Reviewer #2: Yes

5. Review Comments to the Author

Reviewer #1: The manuscript by Acosta et al explores the microbial community of duckweed aquatic plant and shows that the microbial assemblage is pretty constant regardless of the site, plant host, or time of sampling alluding to a "core" duckweed community. Results are convincing for the most part. My problems are with the way the results were presented. The authors were sometimes needlessly descriptive. This is not to say that the work is insignificant. Quite the contrary in fact. It just needs to be presented in a more engaging way. I try to give an overview of what I mean in the points below.

1. Overall, numbers of reads, sequences, ASVs, while important, might be better off presented in a table and referred to often rather than mentioned in the text. These seem to interrupt the flow of sentences when in text.

2. When appropriate, the taxonomy of the ASVs encountered is more important to the reader than their number. For example, in lines 406 to 434, the identity of these genera enriched would have been more interesting to know than their numbers. Also, when doing that, a brief discussion of why these genera could possibly be enriched in the duckweed community compared to the surrounding ambient water is needed. The authors did some of that in the discussion section but having this in the results text would make this section more reader-friendly.

3. I was a little confused on the difference between PM wastewater studies 1 and 2. The rationale behind doing this twice is a little unclear, as was the rationale on treating the leaves the second time around but not the first time.

4. Also, referring to the PM wastewater studies, instead of calling the different communities "sample types" maybe a better way of presenting these comparison would be initial community (duckweed pre-enrichment) versus final community (duckweed post incubation with wastewater and this community here can be further subdivided to different incubation times) versus the innoculum community (which is the wastewater).

5. An important point that I found missing with the "core" duckweed microbiome is whether the authors had a preset cutoff for the abundance of genera that are deemed "core". For example, did the authors decide that a genus is core if it occurs in all samples even if this occurrence was rare? If there was a preset arbitrary abundance cutoff the authors used (I think there should be one), they should mention that.

6. Also, it would be easier if the authors set a cutoff for what they consider "noise". Anything that does not pass this cutoff should not be discussed in the text. For example, if a noise cutoff is set, lines 502-504 should be removed from text and so on.

Reviewer #2: Review of PONE-D-19-25118, Acosta et al., Duckweed hosts a taxonomically similar bacterial assemblage as the terrestrial leaf microbiome.

Overall, this is an interesting and useful study, and is worthy of publication in PLoSOne. The experiments are relatively straightforward, the results solid, and the interpretation and discussion are sound and do not speculate beyond what is justified by the data. It is fascinating that these greatly reduced aquatic angiosperms have similar microbiomes to terrestrial plants, implying some universal plant microbiome assembly “rules.” Note that I am not an expert in the methods used to characterize the microbiome, so cannot critically evaluate those aspects of the study.

My main issue is that the writing style was laborious to read and made it difficult to follow the meaning of many statements. This is especially critical to help the reader, given the complex and abstract statistical analyses of the results. I strongly recommend going through the entire MS and rewriting in active voice, using more direct wording, and shortening sentences. I made a number of suggestions directly in the MS (uploaded), but these are only examples and are not comprehensive. There are also numerous places that probably require commas (some indicated in MS), some of which may be obviated by rewording. All genus/species names need to be italicized. There are various places (only a few of which are indicated in the uploaded MS) that seem like they need a reference cited.

I only have a couple substantive comments:

Discussion on pp. 24-25: Unlike much other vegetation, duckweeds reproduce predominantly by asexual vegetative means. This means physical continuous contact between parent and progeny that might enable direct transfer of bacteria between individuals, as opposed to new individuals starting life physically separate from the parent via seeds. It may be worth briefly discussing this.

Lines 520-522: It sounds like it may be worth stating more explicitly that the core genera were not necessarily the most abundant, if I am interpreting these statements correctly.

6. PLOS authors have the option to publish the peer review history of their article (what does this mean?). If published, this will include your full peer review and any attached files.

Reviewer #1: No

Reviewer #2: No

---

## [Author Response · Author response to Decision Letter 0]

7 Jan 2020

Academic Editor

To enhance the reproducibility of your results, we recommend that if applicable you deposit your laboratory protocols in protocols.io, where a protocol can be assigned its own identifier (DOI) such that it can be cited independently in the future. For instructions see: http://journals.plos.org/plosone/s/submission-guidelines#loc-laboratory-protocols

- Appropriate protocols have been uploaded to protocols.io (dx.doi.org/10.17504/protocols.io.98zh9x6).

- Headings were re-written in sentence case.

- Figure citations and captions were re-formatted.

- A statement was added to manuscript indicating written permission was granted by United Water Princeton Meadows Inc in a written agreement with Rutgers University to use tertiary wastewater samples from the treatment plant for academic studies.

3. We note that Figure 1 in your submission contain map/satellite images which may be copyrighted. All PLOS content is published under the Creative Commons Attribution License (CC BY 4.0), which means that the manuscript, images, and Supporting Information files will be freely available online, and any third party is permitted to access, download, copy, distribute, and use these materials in any way, even commercially, with proper attribution. For these reasons, we cannot publish previously copyrighted maps or satellite images created using proprietary data, such as Google software (Google Maps, Street View, and Earth). For more information, see our copyright guidelines: http://journals.plos.org/plosone/s/licenses-and-copyright.

- The images were removed.

Reviewer #1

1. Overall, numbers of reads, sequences, ASVs, while important, might be better off presented in a table and referred to often rather than mentioned in the text. These seem to interrupt the flow of sentences when in text.

- In the original manuscript, many statistics were embedded in the text. In the revised manuscript, these statistics were moved into supplementary files and/or into figure citations. 

2. When appropriate, the taxonomy of the ASVs encountered is more important to the reader than their number. For example, in lines 406 to 434, the identity of these genera enriched would have been more interesting to know than their numbers. Also, when doing that, a brief discussion of why these genera could possibly be enriched in the duckweed community compared to the surrounding ambient water is needed. The authors did some of that in the discussion section but having this in the results text would make this section more reader-friendly.

- Statistics embedded in text were removed where appropriate. More detailed descriptions for the observed OTUs were written throughout the revised manuscript.

3. I was a little confused on the difference between PM wastewater studies 1 and 2. The rationale behind doing this twice is a little unclear, as was the rationale on treating the leaves the second time around but not the first time.

- The first study compared the DAB community against wastewater communities. The second study compared the DAB community between different duckweed species and tissue treatments.

This was explained more concisely and explicitly in the revised manuscript.

4. Also, referring to the PM wastewater studies, instead of calling the different communities "sample types" maybe a better way of presenting these comparison would be initial community (duckweed pre-enrichment) versus final community (duckweed post incubation with wastewater and this community here can be further subdivided to different incubation times) versus the inoculum community (which is the wastewater).

- In the initial manuscript, the word “sample” was used to refer to bacterial communities. In the revised manuscript, bacterial communities are referred to as bacterial communities. As per the reviewers’ suggestion, acronyms were used to help distinguish the different communities between the different studies. 

5. An important point that I found missing with the "core" duckweed microbiome is whether the authors had a preset cutoff for the abundance of genera that are deemed "core". For example, did the authors decide that a genus is core if it occurs in all samples even if this occurrence was rare? If there was a preset arbitrary abundance cutoff the authors used (I think there should be one), they should mention that.

- In the initial manuscript the core microbiome was defined as those taxa that were found in each study. No abundance cutoff was set, thus the minimal number of reads was “1”. In the revised manuscript, core taxa were first distinguished by their presence in most of the studies. This group of core taxa was further refined by only including those taxa whose abundance was 2-fold greater than the average community abundance for each study.

6. Also, it would be easier if the authors set a cutoff for what they consider "noise". Anything that does not pass this cutoff should not be discussed in the text. For example, if a noise cutoff is set, lines 502-504 should be removed from text and so on.

- It can be difficult to distinguish between noise (technical) and rare taxa (biological) but easier to determine significant versus insignificant. Differential abundance testing and beta-diversity statistics were used to verify significant differences between communities.

Reviewer #2

My main issue is that the writing style was laborious to read and made it difficult to follow the meaning of many statements. This is especially critical to help the reader, given the complex and abstract statistical analyses of the results. I strongly recommend going through the entire MS and rewriting in active voice, using more direct wording, and shortening sentences. I made a number of suggestions directly in the MS (uploaded), but these are only examples and are not comprehensive. There are also numerous places that probably require commas (some indicated in MS), some of which may be obviated by rewording. 

- Passive voice was replaced with active voice. Fuzzy verbs were replaced with action verbs. Sentence structure was modified to clarify the sentence meaning.

All genus/species names need to be italicized. 

- Genus names (e.g.: Arabidopsis) were italicized. Arabidopsis thaliana was italicized in first mention but referred to as Arabidopsis thereafter.

There are various places (only a few of which are indicated in the uploaded MS) that seem like they need a reference cited.

- Missing references were added to Introduction and Discussion section. 

Discussion on pp. 24-25: Unlike much other vegetation, duckweeds reproduce predominantly by asexual vegetative means. This means physical continuous contact between parent and progeny that might enable direct transfer of bacteria between individuals, as opposed to new individuals starting life physically separate from the parent via seeds. It may be worth briefly discussing this.

- This was used to explain duckweed microbiome stability in the Discussion section.

Lines 520-522: It sounds like it may be worth stating more explicitly that the core genera were not necessarily the most abundant, if I am interpreting these statements correctly.

- In the revised manuscript, Figure 7 addresses core taxa abundance and how it relates to the rest of the community. Core taxa were re-defined as taxa which contained an abundance 2 times greater than the rest of the community in at least 6 of the 7 studies.

---

## [Decision Letter · Decision Letter 1]

21 Jan 2020

Duckweed hosts a taxonomically similar bacterial assemblage as the terrestrial leaf microbiome

PONE-D-19-25118R1

Dear Dr. Lam,

We are pleased to inform you that your manuscript has been judged scientifically suitable for publication and will be formally accepted for publication once it complies with all outstanding technical requirements.

With kind regards,

Ulrich Melcher

Academic Editor

PLOS ONE

Additional Editor Comments (optional):

Reviewers' comments:

Reviewer's Responses to Questions

**Comments to the Author**

1. If the authors have adequately addressed your comments raised in a previous round of review and you feel that this manuscript is now acceptable for publication, you may indicate that here to bypass the “Comments to the Author” section, enter your conflict of interest statement in the “Confidential to Editor” section, and submit your "Accept" recommendation.

Reviewer #1: (No Response)

Reviewer #2: All comments have been addressed

2. Is the manuscript technically sound, and do the data support the conclusions?

Reviewer #1: Yes

Reviewer #2: Yes

3. Has the statistical analysis been performed appropriately and rigorously? 

Reviewer #1: Yes

Reviewer #2: Yes

4. Have the authors made all data underlying the findings in their manuscript fully available?

Reviewer #1: Yes

Reviewer #2: Yes

5. Is the manuscript presented in an intelligible fashion and written in standard English?

Reviewer #1: Yes

Reviewer #2: Yes

6. Review Comments to the Author

Reviewer #1: The authors addressed all my comments from the first round of reviews. Please make sure to italicize all bacterial genera names in the manuscript.

Reviewer #2: (No Response)

7. PLOS authors have the option to publish the peer review history of their article (what does this mean?). If published, this will include your full peer review and any attached files.

Reviewer #1: No

Reviewer #2: No

---

## [Editor Report · Acceptance letter]

29 Jan 2020

PONE-D-19-25118R1 

Duckweed hosts a taxonomically similar bacterial assemblage as the terrestrial leaf microbiome 

Dear Dr. Lam:

I am pleased to inform you that your manuscript has been deemed suitable for publication in PLOS ONE. Congratulations! Your manuscript is now with our production department. 

With kind regards,

on behalf of

Dr. Ulrich Melcher 

Academic Editor

PLOS ONE